# Prioritising Risk Factors for Prescription Drug Overdose among Older Adults in South Korea: A Multi-Method Study

**DOI:** 10.3390/ijerph18115948

**Published:** 2021-06-01

**Authors:** Eun-Hae Lee, Ju-Ok Park, Joon-Pil Cho, Choung-Ah Lee

**Affiliations:** 1Division of Injury Prevention and Control, Korea Disease Control and Prevention Agency, Cheongju-si 28159, Korea; leeeunhae7@naver.com; 2Department of Emergency Medicine, Hallym University, Dongtan Sacred Heart Hospital, Hwaseong-si 18450, Korea; juok.park@gmail.com; 3Department of Emergency Medicine, Ajou University School of Medicine, Suwon 16499, Korea; jpcho6007@gmail.com

**Keywords:** prescription drugs, risk factors, polypharmacy

## Abstract

Older adults are vulnerable to drug overdose. We used a multi-method approach to prioritise risk factors for prescription drug overdose among older adults. The study was conducted in two stages. First, risk factors for drug overdose were classified according to importance and changeability through literature review, determined through 2-phase expert surveys. Second, prescription drug overdose cases during 2011–2015 were selected from a national cohort; the prevalence of ‘more important’ or ‘more changeable’ factors determined in stage one was investigated. Scores were assigned according to the Basic Priority Rating Scale formula, reflecting the problem size and seriousness and intervention effectiveness. In the first stage, polypharmacy, old-old age, female sex, chronic disease, psychiatric disease, and low socioeconomic status (SES) were selected as risk factors. In the second stage, 93.9% of cases enrolled had chronic medical disease; 78.3% were using multiple drugs. Low SES was more prevalent than other risk factors. As per the scoring formula, chronic medical disease, polypharmacy, psychiatric disease, low SES, female sex, and old-old age were the most important risk factors in order of priority. Patients with chronic medical disease and those using multiple medications should be prioritised in overdose prevention interventions among older adults.

## 1. Introduction

Deaths from drug overdose remain a significant public health concern [1]. Although programmes for prevention are being conducted and monitored, drug overdose remains the leading cause of injury-related deaths. Indeed, in the United States, mortalities attributed to drug overdose have been steadily increasing by an average of 16% per year, between 2014 and 2017 [2,3]. The National Office for Statistics in the UK reported that the mortality rate from drug misuse has been increasing since 1993 and was estimated to be 50.4 deaths per million people in 2019 [4]. Because Korea lacks a poisoning information management centre, it is difficult to obtain accurate epidemiological statistics. According to the National Emergency Department Information System of the national emergency medical centre, the hospitalisation rate of patients with drug overdose is increasing [5]. Approximately 50,000 patients visit the emergency room owing to acute overdose, and pharmaceutical drugs account for the largest proportion at 44.2% [6].

Drug overdose can occur at any age, but older adults are particularly vulnerable. Older people have a high prevalence of multiple chronic medical conditions; therefore, they use more prescription drugs than those in other age groups. At an older age, the proportion of body fat is relatively higher than that of skeletal muscle, resulting in an increase in the volume of distribution. Ageing is associated with a reduction in first-pass metabolism owing to a reduction in liver mass and blood flow [7]. Some diseases also affect drug pharmacokinetic changes with ageing. The heart rate is decreased and systemic vascular resistance is increased in older patients with congestive heart failure. Anticoagulants, psychotropics, diuretics, cardiovascular and respiratory drugs, and other drugs cause pharmacodynamic changes with ageing [8]. Drug clearance decreases with the occurrence of ageing-related renal and hepatic function impairment [9]. Moreover, older people are more likely to be exposed to multiple drugs owing to comorbidities, and the risk of drug–drug interaction also increases as a result of age-related changes in drug pharmacokinetics and pharmacodynamics [10]. Older adults are therefore at a high risk for mortality from overdose.

To prevent overdose among older adults, it is important to identify risk factors that are specific to this age group and prioritise them to effectively proceed with an intervention. Priority setting determines the strategic directions of the national health plan. As resources are always limited, priority setting should reflect not only medical factors such as the incidence and severity of disease but also other factors such as cost-effectiveness, societal values, and the needs of stakeholders. Prioritisation not only helps to make the most of financial resources but also helps meet the needs of the community. Prioritising tools include the multi-voting technique, prioritisation matrix, nominal group technique, Hanlon method, and Basic Priority Rating Scale (BPRS) [11].

In this study, we aimed to prioritise the risk factors for prescription drug overdose among older adults using the BPRS.

## 2. Materials and Methods

A multi-method design based on the BPRS concept was applied in this study [12]. It was performed in two stages. In the first stage, primary risk factors were selected by conducting a literature review on risk factors for prescription drug overdose among older adults. A two-phase modified Delphi was performed for the primary selection of risk factors [13]. In the second stage, the final priority was calculated for the risk factors that were initially selected using the BPRS formula (Figure 1) [14].

PATCH, Planned Approach to Community Health; BPRS, Basic Priority Rating Scale; PEARL, propriety, economics, acceptability, resources, legality.

### 2.1. Stage 1: Primary Selection of Target Risk Factors for Prioritising

We conducted a literature review for known risk factors of drug overdose. Drug overdose cases were defined as all those resulting from drug misuse, drug abuse, and overuse of a drug for medical reasons. Both unintentional and intentional overdose were included; cases of adverse drug events or hypersensitivity were excluded. The first Delphi survey was conducted to select primary items for reviewed risk factors. An expert survey was conducted online in two phases. The phase two questionnaire was repeatedly evaluated and surveyed by experts who responded in phase one. The results obtained through the first phase were analysed and sent back to all respondents to inform them of other experts’ opinions and to accommodate a readjustment of their opinions in the second phase. 

#### 2.1.1. Expert Panel

Thirty scientific members comprising 23 directors of the Emergency Department-based Injury In-depth Surveillance and 7 experts working in the field of injury prevention for at least >5 years were selected. Thereafter, the experts were invited to participate in this study through an email stating the aim of the study and survey questionnaire. In total, 18 scientific members comprising 15 directors and 3 session experts agreed to participate.

#### 2.1.2. Definition of Agreement on Delphi

The Delphi questionnaire was designed based on literature review. According to the Planned Approach to Community Health concept managed and supported by the Centers for Disease Control, we utilised two criteria—importance and changeability [15]. Regarding the importance, the question ‘How important is each risk factor for affecting older adults’ overdose in their personal health and community was answered with ‘more important’ or ‘less important’. For changeability, participants were asked to answer the question ‘how easy or difficult will it be to intervene in the issue by controlling each risk factor?’ with ‘more changeable’ or ‘less changeable’. Less important and less changeable factors were not prioritised and excluded from the next analysis. Agreement for importance was achieved when more than two-thirds of experts considered a factor to be ‘more important’. Agreement for changeability was achieved when more than half of the expert panel deemed a factor to be ‘more changeable’. Finally, factors classified as ‘more important’ or ‘more changeable’ were selected, and further investigation was conducted in stage two. A pretest was conducted by two researchers with master’s degrees in public health, and two health research experts reviewed the content validity.

### 2.2. Stage 2: Prioritisation by the BPRS

The BPRS was applied to prioritise the risk factors [14]. The BPRS prioritises health problems or risk factors based on the nature of the problem (defined by size and seriousness) and effectiveness of the solution.

In the first step, scores were calculated for each element using the following formula.
$$BPRS=\frac{\;\left(A+B\;\right)C}{3}$$
*A*, size of the problem; *B*, seriousness of the problem; *C*, effectiveness of intervention.

For the second step, the propriety, economics, acceptability, resources, and legality (PEARL) test was applied. The PEARL test is used to eliminate any factors that receive an answer of ‘No’ to any of the questions on aspects of feasibility. Finally, according to the result value calculated by risk factor, the factor with the highest score was ranked first, followed by others in that order [16].

#### 2.2.1. Data Sources

The National Health Insurance Service-National Sample Cohort (NHIS-NSC) [17] and expert survey were used as a data source for the BPRS calculation. The NHIS-NSC is a population-based cohort established by the National Health Insurance Service in South Korea. The NHIS-NSC includes detailed information such as the patient’s age, sex, place of residence, ICD-10 based diagnosis, treatment dates, procedure/operation, type and quantity of prescription drugs, and medical expenses for all health insurance holders. In fact, it is a large-scale database that can verify the long-term effectiveness of an intervention in a prospective cohort [18]. Patients who had prescription drug overdose between 2011 and 2015 were selected from this cohort. Among 5,222,094 cases of medical institution use during the period, patients who visited the emergency room with prescription drug overdose codes were identified (Appendix A). We set up a one-month wash-out period.

#### 2.2.2. Variable

A group of patients with risk factors identified through literature review was defined. Old-old age was defined as an age over 85 years [19]. The population with chronic disease was defined by the presence of one or more medical records pertaining to hypertension, diabetes mellitus, heart disease, malignancy, chronic renal disease, and cerebrovascular disease in at least 5 years prior to the beginning of the case-defining period. Low SES was defined by inclusion in the lowest 10% of health insurance premiums determined by income. Psychiatric disease was defined by the presence of a medical history according to the F code for at least 5 years prior to the beginning of the case-defining period. Polypharmacy was defined by the intake of two or more drugs. If the drug was prescribed within 1 year prior to visiting the medical institution owing to drug overdose, the last date of suspected drug usage could be estimated by adding the number of prescribed days from the date of prescription. When the last date of suspected drug usage was the same as or followed the date of visit with drug overdose, the patients were presumed to have been currently taking the drug.

#### 2.2.3. Definition of Criterion and Scores

The size of a problem (criterion A) is most often represented by incidence or prevalence rates in a 100,000 population. To determine the size of the risk factor, we modified it such that the etiologic fraction was calculated for each factor. The seriousness of the problem (criterion B) was composed of four sub-criteria: urgency, severity, economic loss, and impact on other people. Each sub-criterion was scored on a scale of 0 to 5, and the total score for the seriousness component was 20 points. Urgency was originally defined as the degree of emergent nature, which required a rapid response to prevent the spread of the problems or death. We calculated the risk ratio of death among overdosed patients with risk factors for each 5-year period, and classified this trend into ‘decreasing’, ‘stabilising’, or ‘increasing’; we then assigned 1, 3, and 5 points, respectively. The severity, defined as the fatality rate, was modified with the intensive care unit admission rate to make deviations for each risk factor. Economic loss is the accumulation of costs associated with the health problem and that borne by society. In this study, only the direct cost owing to prescription drug overdose of patients with this factor was reflected. Scores for impact on other people, effectiveness of intervention (criterion C), and PEARL were included in the second Delphi in two phases. The questionnaire items of phases one and two were configured identically. The second Delphi survey was conducted to allow participants to modify their response results, considering the responses of other study participants in the first survey. The impact on others was evaluated with a full score of 5, and the feasibility of the intervention was evaluated with a full score of 10. NHIS-NSC analysis results were not provided to avoid any influence on the subjective decision of experts. The expert panel was structured identical to that in the first Delphi (Table 1). 

### 2.3. Ethics

Ethical approval was obtained from the Institutional Review Board of the Ajou University (Approval number: AJIRB-SBR-EXP-19-104) and was performed in accordance with the principles embodied in the Declaration of Helsinki. Written informed consent was obtained from each participant.

## 3. Results

Through the literature review, the risk factors of prescription drug overdose in older adults were identified to be concurrent chronic medical disease, polypharmacy, concurrent psychiatric illness, female sex, old-old age, social isolation, physical disabilities/reduced mobility, chronic pain, history of alcohol problems, and transitions in care/living situations [20,21,22,23,24]. Chronic medical diseases included heart disease, chronic lung disease, diabetes, chronic kidney disease, and cerebrovascular disease. Musculoskeletal disease was classified into the category of physical disabilities/reduced mobility, and psychiatric disorders, including Alzheimer’s disease, were classified separately. 

Among the 30 contacted experts, 18 (60%) agreed to participate in the study. All responders completed all rounds; respondent characteristics are shown in Table 2. Experts ranked polypharmacy as ‘more important’ and ‘more changeable’ and answered that old-old age, female sex, existing chronic medical and psychiatric disease, and low socioeconomic status (SES) were important but less changeable issues (Table 3). The BPRS scores were calculated for the six risk factors that were finally selected. 

Between 2011 and 2015, the number of cases enrolled in the sample cohort was 5,222,094. Among them, 525,580 cases were treated in the emergency room, 1506 were treated due to prescription drug overdose, and 327 were over 65 years old (Figure 2). In order to evaluate the urgency, the mortality rate was evaluated in patients with each risk factor among the total number of patients by year (Figure 3).

Table 4 shows the calculated indicators of the six risk factors. Among all the overdosed patients, 93.9% had chronic medical disease, and 78.3% were using multiple drugs. Among the patients admitted to the intensive care unit whose severity was measured, 94.9% had chronic medical disease. The proportion of patients with low SES was greater than that of those with other risk factors.

All experts who participated in the first Delphi also participated in the second one. Experts provided the highest score of five for the impact of low SES and chronic disease in preference to others. In addition, their answers indicated that intervention for the patient group using multiple medications was the most effective. For all risk factors, one point was awarded on the PEARL score.

According to the BPRS, the presence of chronic medical disease was ranked as the most important risk factor with 78 points, followed by polypharmacy with 70 points. Both psychiatric disease and low SES were placed in the same ranking with 45 points; female sex had 35 points and old-old age 21 points (Table 5).

## 4. Discussion

This multi-method study suggested prioritising the known risk factors of drug overdose in older adults to establish prevention strategies. Concurrent chronic medical disease emerged as the highest priority risk factor, followed by polypharmacy, concurrent psychiatric disease, low SES, female sex, and old-old age.

Risk factors for drug overdose in the elderly are well known. However, previous studies showed risk factors only considering causality. In this study, we proposed prioritisation by comprehensive consideration of social resources, social influence, effectiveness of intervention, and economic costs. Most communities do not have enough resources to solve all health problems and target groups at once. Therefore, we need to prioritise problems or causes and plan to address some of them first and some later. Prioritisation is an important component of a systematic plan of health promotion; it also influences the development of a comprehensive assessment plan [15]. Prioritising tools can be categorised into subjective and objective approaches. The former includes the simplex method, nominal group technique, and multi-voting technique, and the latter includes Delphi, prioritisation matrix, and BPRS [11]. The BPRS offers the advantage of having predetermined criteria, standardised comparisons, and the use of a rubric that minimises bias; it has therefore been widely used in health settings. This requires the use of predetermined criteria, standardised comparisons, and the use of a rubric that minimises bias. One model that incorporates these criteria and has a relatively long history of use in health settings is the BPR model.

Multi-method research is a study type that uses data from one or more sources and different types of analysis [25]. A multi-method approach can combine different data sources, methods, or observers to validate data and results, discover fresh elements that stimulate further work, and widen the scope of the study to take in contextual aspects of the situation [26]. In this study, the subjective opinions of stakeholders and experts, and analysis of national sample data were combined to select a prioritising method called BPRS. 

Chronic medical disease was shown to be the most important risk factor for prescription drug overdose in older adults, and the size and severity of the problem were the factors that affected the high score the most. It was found that 93.9% of patients with drug overdose in the emergency department and 94.9% of patients admitted to the intensive care unit had at least one chronic condition. This contributed to chronic medical disease having the highest score among all risk factors. The prevalence of chronic disease increases with age and, therefore, the number of chronic diseases in older adults is inevitably high. Furthermore, recent studies reported that adverse drug reactions were estimated to cause 10–20% of hospital admissions in older adults [27,28].

The prevalence and seriousness of chronic medical disease and polypharmacy are correlated; this causes older people to be more vulnerable to acute overdose and its associated consequences [29]. Polypharmacy in older adults has already been raised as a significant public health problem. In a study that conducted in-home interviews, 87% of older adults were using medications and 36% used five or more drugs simultaneously [30]. It is estimated that 50% of Medicare beneficiaries receive 5 or more drugs [31]. In a study on ambulatory senior adults with cancer, 84% and 43% of the subjects received 5 and 10 or more drugs, respectively [32].

We speculated that the size and severity of polypharmacy problems would be similar to those of chronic medical disease, but the former had lower scores. The use of over-the-counter drugs, herbal medicine, and dietary supplements could not be confirmed in this data source, and it is believed that cases of drug overdose caused by ingesting other people’s medicine could not be identified. Despite the possibility that the size and seriousness of the problem may have been underestimated, the reason polypharmacy was ranked second is because experts considered the effectiveness of the intervention that exerts the greatest influence on BPRS priorities to be high. It is believed that this is because the occurrence of chronic diseases is difficult to control, but prescription and management of the corresponding drugs are judged to be more effective. Interventions targeting polypharmacy are already in progress. It is recommended to reduce the number of drugs or minimise prophylactic drugs in the drug prescription stage [33]. Precautions may be taken regarding drug–drug interactions or adverse drug events [34]. Although scientific evidence is insufficient, drug prescription and administration are computerised [35]. 

Drug-overdosed patients with psychiatric disease constituted 76.1% of the total overdosed cases, which indicates a significant problem. This result was comparable to those of previous studies that showed high correlation between overdose and psychiatric disease [36]. In addition to intentional overdose due to drug dependence, non-intentional overdose is caused by ageing-related diseases such as dementia. Moreover, the hospitalisation rate due to non-intentional overdose of dementia patients is twice as high as that of non-dementia patients [37]. Regarding psychiatric disease, the size and severity of the problem were as high as those in previous studies, but the mortality tended to decrease; the priority was therefore lower than that of other risk factors.

Low SES, one of the risk factors, was investigated to have a smaller problem size despite the fact that the score in the seriousness criterion was second only to chronic medical disease. As low SES was defined by inclusion in the lowest 10% of the income level, the true size of the problem could not be evaluated. This revealed a limitation of BPRS, in that the priority of minority groups could be inadequately evaluated.

Females consume more types and doses of medicine than males and are known to experience more adverse reactions [32]. In this study, the proportions of women among overdosed patients and among those admitted to the intensive care unit were slightly higher. However, there was stability in the mortality trend, which corroborates similar findings reported in other studies during the same period [38]. The scores for the effects of interventions and their impact on others or women were low; therefore, female sex was ranked fifth in the priority list. 

As with low SES, the size of the problem posed by old-old age was small because the study included few super-aged persons. Although urgency in the old-old age population did not diminish, the expert survey regarding the effectiveness of intervention was negative, causing this factor to remain at the bottom of the relative priority. 

There are several limitations of this study. First, NHIS-NSC data are based on billing data for costs incurred during treatment and are not developed for clinical research. Since cases were selected only using the ICD-10 codes, those related to potential overdose, chronic overdose, or adverse drug effects may have been omitted. In addition, although over-the-counter drugs and herbal and dietary supplements comprise a large proportion of the substances causing overdose in older adults, they were not included in this study. Second, we failed to consider disease burden such as disability-adjusted or quality-adjusted life years in the economic cost; instead, the exact direct medical expenses were applied. Finally, the expert group selected to evaluate the effectiveness of intervention, which is the most important criterion of the BPRS, is biased. It was not possible to include practitioners who performed the intervention and the geriatricians as experts. Instead, an advisory group on the prevention of elderly injuries was included in the panel.

## 5. Conclusions

In conclusion, patients with chronic medical disease and those using multiple medications need to be given priority in intervention policies to prevent drug overdose in older adults. In addition, healthcare services should be determined by integrating ethical factors such as human rights, self-determination, equality, and justice as well as cost, effectiveness, and availability of services. In the future, it will also be necessary to develop a prioritising method that considers minority groups, for whom the size of the problem may have been underestimated.

## Figures and Tables

**Figure 1 ijerph-18-05948-f001:**
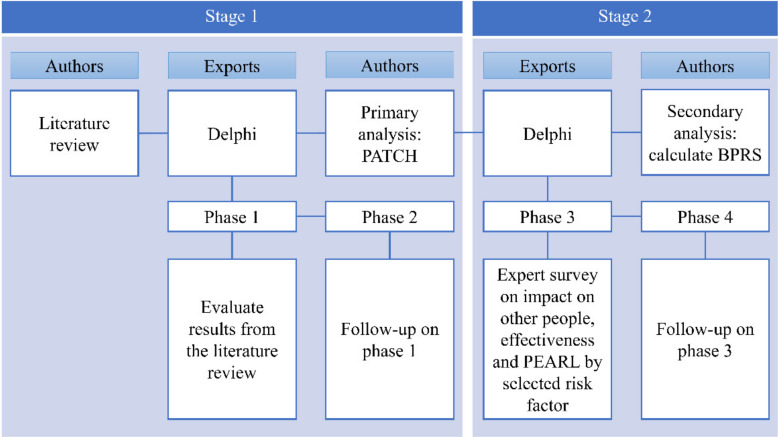
Summary of methods.

**Figure 2 ijerph-18-05948-f002:**
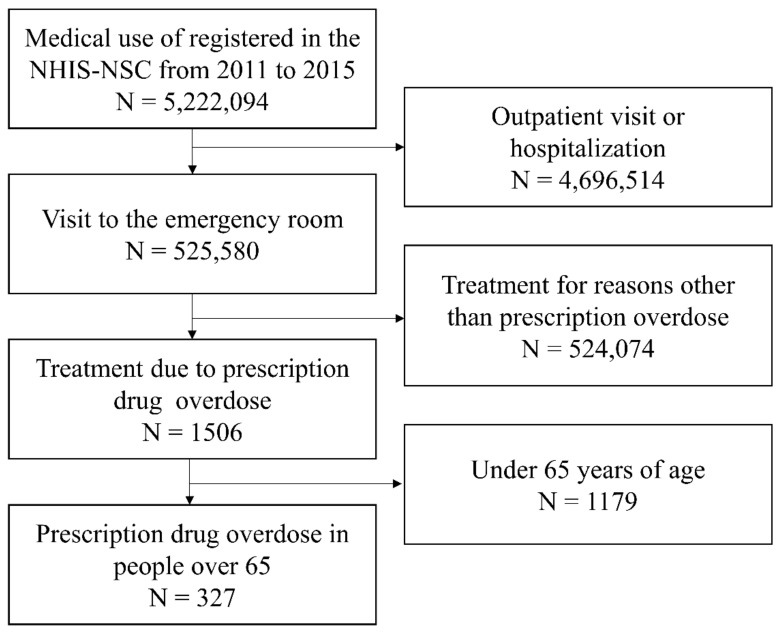
Flowchart of sample selection process.

**Figure 3 ijerph-18-05948-f003:**
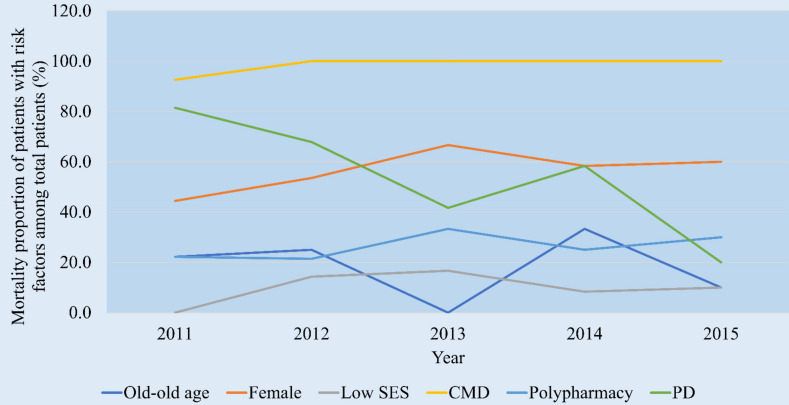
Trend of mortality proportion in overdosed patients with risk factors. SES, socioeconomic status; CMD, chronic medical disease; PD, psychiatric disease.

**Table 1 ijerph-18-05948-t001:** Definition of criteria and scores.

	Size		Seriousness								Effectiveness of Intervention	
			Urgency		Severity		Economic Cost		Impact on Others			
Indicator	Proportion of patients with drug overdose who have the relevant risk factors (%)		Trend of 5-year mortality in patients with risk factors		Admission rate to the intensive care unit for patients with this risk factor (%)		Medical expenses owing to drug overdose in patients with the relevant risk factors (1000 won)		Expert opinion (Likert 5 scale)		Expert opinion (score out of 10)	
Score range	1–10		1–5		1–5		1–5		1–5		1–10	
Distribution of score	~10 10~20 20~30 30~40 40~50 50~60 60~70 70~80 80~90 90~100	1 2 3 4 5 6 7 8 9 10	Decreasing Stabilising Increasing	1 3 5	20 20–40 40–60 60–80 80–100	1 2 3 4 5	<1000 1000–1200 1200–1400 1400–1600 >1600	1 2 3 4 5	No affect Minor affect Neutral Moderate affect Major affect	1 2 3 4 5	No effective Neutral Extremely effective	1 2 3 4 5 6 7 8 9 10
Data source	NHIS-NSC		NHIS-NSC		NHIS-NSC		NHIS-NSC		Delphi		Delphi	

**Table 2 ijerph-18-05948-t002:** Demographics of participants on the expert panel.

	N (%)
Sex	
Female	4 (22.2)
Male	14 (77.8)
Age (years), median (IQR)	44.5 (38–57)
Professions	
Emergency physician	15 (83.3)
Professor in the injury prevention research institute	3 (16.7)
Career in injury surveillance and prevention (year), median (IQR)	10 (5–25)

IQR, interquartile range.

**Table 3 ijerph-18-05948-t003:** Categorisation of risk factors selected by the first expert survey.

	Setting Priorities	
	More Important	Less Important
More changeable	Polypharmacy	
Less changeable	Old-old age Female Chronic medical disease Low SES Psychiatric disease	Social isolation Physical disabilities/reduced mobility Chronic pain History of alcohol problem Transitions in care/living situations

SES, socioeconomic status.

**Table 4 ijerph-18-05948-t004:** Data elements associated with the Basic Priority Rating Scale.

	Size (%)	Seriousness				Effectiveness of Intervention (Median, IQR)
		Urgency	Severity (%)	Economic Cost (won)	Impact on Others (Median, IQR)	
Old-old	12.8	Stabilising	16.2	1,396,531	1 (1–3)	7 (5–9)
Female	56.6	Stabilising	51.5	1,384,519	3 (1–3)	6 (3–6)
Low SES	10.1	Increasing	4.0	1,647,413	5 (3–5)	8 (3–9)
Chronic medical disease	93.9	Stabilising	94.9	1,446,485	5 (1–5)	9 (7–10)
Polypharmacy	78.3	Stabilising	72.7	1,353,738	4 (3–5)	10 (7–10)
Psychiatric disease	76.1	Decreasing	70.7	1,135,366	3 (1–5)	8 (6–10)

SES, socioeconomic status; IQR, interquartile range.

**Table 5 ijerph-18-05948-t005:** Prioritisation of risk factor using the Basic Priority Rating Scale.

Risk Factor	Size	Seriousness	Effectiveness of Intervention	P	E	A	R	L	Total BPRS	Rank
Chronic medical disease	9	3 + 5 + 4 + 5 = 17	9	1	1	1	1	1	78	1
Polypharmacy	7	3 + 4 + 3 + 4 = 14	10	1	1	1	1	1	70	2
Psychiatric disease	7	1 + 4 + 2 + 3 = 10	8	1	1	1	1	1	45	3
Low SES	1	5 + 1 + 5 + 5 = 16	8	1	1	1	1	1	45	3
Female	5	3 + 3 + 3 + 3 = 12	6	1	1	1	1	1	34	5
Old-old age	1	3 + 1 + 3 + 1 = 8	7	1	1	1	1	1	21	6

PEARL, propriety, economics, acceptability, resources, legality; BPRS, Basic Priority Rating Scale; SES, socioeconomic status.

## Data Availability

The data used to support the findings of this study are available from the corresponding author upon request.

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
