# Peer review of "Prioritising Risk Factors for Prescription Drug Overdose among Older Adults in South Korea: A Multi-Method Study"

_ijerph, 2021, doi:10.3390/ijerph18115948_

Round 1

Reviewer 1 Report

The authors present a work regarding putting in order risk factors that could be related with drug poisoning for older adults. It would be better if we could have a definition or description to what is drug poisoning in this work. Intended? Unintended? Did the study exclude suicidal cases? Overdose? Fatal adverse drug event? Wrong drug administration? Lines 46-53. It would be better to explain how the Pharmacokinetic parameters are associated with poisoning, I guess refer to overdose? Generally, introduction should provide some additional pharmacological and epidemiological data. Only body fat and kidney function play role? Do all countries have the same issue? Can study results be extrapolated globally? Is it about Korea? Are there any similar works with related outcomes? I believe since it enrolls S. Korea population, it should be reflected also in the title and explained in detail the aim of the study. Moreover, the study results lack of novelty, they present more-or-less the same issues that are known to be associated with drug poisoning. Was the prioritization from experts’ group in accordance with the data analysis? What new do we learn from this study? Line 289, are there non "therapeutic" drugs? Which of the reasons described here could be related with adverse drug reactions and which could be adverse drug events? Study bias from experts’ part? Overall the manuscripts requires improvements and especially if it is intended to be further exploited to plan healthcare strategies in S. Korea (as they describe in discussion).

Author Response

We sincerely appreciate the careful review of our manuscript and the helpful suggestions provided, which have contributed considerably to its improvement. The manuscript has been revised as suggested, and detailed responses to the individual comments have been enclosed.

1. The authors present a work regarding putting in order risk factors that could be related with drug poisoning for older adults. It would be better if we could have a definition or description to what is drug poisoning in this work. Intended? Unintended? Did the study exclude suicidal cases? Overdose? Fatal adverse drug event? Wrong drug administration?  

→ We appreciate your pertinent observations and questions. We have also deliberated at length on the term ‘drug poisoning’. In this study, the term ‘drug poisoning’ refers to drug overdoses resulting from drug misuse, abuse, and overuse for medical reasons. Both unintentional and intentional overdosing were included; cases of adverse drug events or hypersensitivity were not included. As we believed that the term ‘drug poisoning’ could be confusing, it has been to ‘drug overdose’.

2. Lines 46-53. It would be better to explain how the Pharmacokinetic parameters are associated with poisoning, I guess refer to overdose? Generally, introduction should provide some additional pharmacological and epidemiological data. Only body fat and kidney function play role?  

→ We appreciate your observations. As suggested, we have added further details in the Introduction regarding the effect on pharmacokinetics and pharmacodynamics of drugs in the elderly.

3. Do all countries have the same issue? Can study results be extrapolated globally? Is it about Korea? Are there any similar works with related outcomes? I believe since it enrolls S. Korea population, it should be reflected also in the title and explained in detail the aim of the study.  

→ We appreciate your pertinent questions. The burden of drug overdose in other countries has been mentioned, and statistics from South Korea have been added. The title and aim of the study state that the study was intended for the South Korean population.

4. Was the prioritization from experts’ group in accordance with the data analysis?  

→ NHIS-NSC analysis results were not provided to ensure that the subjective decision of the experts was not influenced. This has been mentioned in the Materials and Methods.

5. What new do we learn from this study?  

→ Risk factors for drug overdose in the elderly are well known. However, previous studies showed risk factors only considering causality. In this study, we proposed prioritisation by comprehensive consideration of social resources, social influence, effectiveness of intervention, and economic costs. This has been emphasised on in the Discussion.

6. Line 289, are there non "therapeutic" drugs? Which of the reasons described here could be related with adverse drug reactions and which could be adverse drug events? Study bias from experts’ part?

→ We appreciate your questions and apologise for the confusion caused by the use of the term ‘drug poisoning’. The term has been changed to ‘drug overdose’, and it has been defined in the Materials and Methods.

Reviewer 2 Report

This study used a multi-method approach to prioritise risk factors for prescription drug poisoning among older adults.
Unfortunately,The results that patients with chronic medical illness and those taking multiple drugs should prioritize addiction prevention interventions among the elderly are within expectations.
That makes it difficult to judge the importance of this paper.
However, clarifying them numerically may be important for taking epidemiological studies to the next step and for proper use of drugs.

There seems to be no major mistake in the statistical method used.
The interpretation of the results also seems reasonable.

Therefore, I have no reason to object to the acceptance of this paper.
However, it would be better if there was an additional about the importance of this paper.

Author Response

We sincerely appreciate the careful review of our manuscript and the helpful suggestions provided, which have contributed considerably to its improvement. The manuscript has been revised as suggested, and detailed responses to the individual comments have been enclosed.

Comment)

It would be better if there was an additional about the importance of this paper.

→ As suggested, the importance of this study has been highlighted in the Discussion.

Risk factors for drug overdose to in the elderly are well known. However, the previous studies showed risk factors only considering only causality. In this study, we proposed prioritisation by comprehensive consideration of with social resources, social influence, effectiveness of intervention, and economic costs. Most communities do not have enough resources to solve all health problems and target groups at once. Therefore, we need to prioritise problems or causes and plan to address some of them first and some later.

Round 2

Reviewer 1 Report

The authors presented an updated version of their manuscript and addressed the issues posed from the initial review. I would recommend also that in their introduction, probably somewhere between lines 53-67, they could mention also the factor of drug-drug interactions and particularly pharmacokinetic DDIs that could lead in "overdose" phenomena. It is implied in the results (as polypharmacy) and stated in discussion but it would be optimal if it would be also in the introduction section since polypharmacy is an identified factor. Polypharmacy is related with the occurrence of clinically significant DDIs that may lead in "overdose" toxicity from altered PK/PD parameters of co-administered medications. Apart of that, the manuscript can be further processed as accepted. 

Author Response

We sincerely appreciate the careful review of our manuscript and the helpful suggestions provided, which have considerably contributed in improving our manuscript. The manuscript has been revised as suggested.

 The authors presented an updated version of their manuscript and addressed the issues posed from the initial review. I would recommend also that in their introduction, probably somewhere between lines 53-67, they could mention also the factor of drug-drug interactions and particularly pharmacokinetic DDIs that could lead in "overdose" phenomena. It is implied in the results (as polypharmacy) and stated in discussion but it would be optimal if it would be also in the introduction section since polypharmacy is an identified factor. Polypharmacy is related with the occurrence of clinically significant DDIs that may lead in "overdose" toxicity from altered PK/PD parameters of co-administered medications. Apart of that, the manuscript can be further processed as accepted.

→ Thank you for your suggestion. Accordingly, in the Introduction, we have mentioned that the risk of overdose in the elderly increases owing to an increase in comorbidities, the corresponding increase in the number of medications to be taken, and ageing-related changes in drug pharmacokinetics and pharmacodynamics.
